Ancient divergence time estimates in Eutropis rugifera support the existence of Pleistocene barriers on the exposed Sunda Shelf

Karin Benjamin R. benkarin@berkeley.edu 1 2
Das Indraneil 3
Jackman Todd R. 1
Bauer Aaron M. 1
1 Department of Biology, Villanova University , Villanova , PA , United States of America
2 Museum of Vertebrate Zoology and Department of Integrative Biology, University of California , Berkeley , CA , United States of America
3 Institute of Biodiversity and Environmental Conservation, Universiti Malaysia Sarawak , Kota Samarahan , Sarawak , Malaysia
Wallis Graham
Electronic publication date: 2017 Oct 27
Publication date: 2017
Volume: 5
Electronic Location ID: e3762
Received 2017 Jul 10; Accepted 2017 Aug 15
Copyright: ©2017 Karin et al.
Copyright year: 2017
Copyright holder: Karin et al.
License: This is an open access article distributed under the terms of the Creative Commons Attribution License, which permits unrestricted use, distribution, reproduction and adaptation in any medium and for any purpose provided that it is properly attributed. For attribution, the original author(s), title, publication source (PeerJ) and either DOI or URL of the article must be cited.
License URL: https://creativecommons.org/licenses/by/4.0/

Keywords: Timetree, Phylogeography, Sundaland, Borneo, Rough-scaled sun skink

Funding: National Science Foundation EF 1241885 Villanova University Ministry of Higher Education IA010200-0708-0007 Financial support was provided by a National Science Foundation grant EF 1241885 (subaward 13-0632) to AMB and the Gerald M. Lemole, M.D. Endowed Chair Fund at Villanova University. Additional financial support was provided by Villanova University through a Biology Department Fellowship and a Graduate Student Fellowship to BRK. ID was supported by a Niche Research Grant by the Ministry of Higher Education, Government of Malaysia (IA010200-0708-0007). The funders had no role in study design, data collection and analysis, decision to publish, or preparation of the manuscript.

==============================
Episodic sea level changes that repeatedly exposed and inundated the Sunda Shelf characterize the Pleistocene. Available evidence points to a more xeric central Sunda Shelf during periods of low sea levels, and despite the broad land connections that persisted during this time, some organisms are assumed to have faced barriers to dispersal between land-masses on the Sunda Shelf. Eutropis rugifera is a secretive, forest adapted scincid lizard that ranges across the Sunda Shelf. In this study, we sequenced one mitochondrial (ND2) and four nuclear (BRCA1, BRCA2, RAG1, and MC1R) markers and generated a time-calibrated phylogeny in BEAST to test whether divergence times between Sundaic populations of E. rugifera occurred during Pleistocene sea-level changes, or if they predate the Pleistocene. We find that E. rugifera shows pre-Pleistocene divergences between populations on different Sundaic land-masses. The earliest divergence within E. rugifera separates the Philippine samples from the Sundaic samples approximately 16 Ma; the Philippine populations thus cannot be considered conspecific with Sundaic congeners. Sundaic populations diverged approximately 6 Ma, and populations within Borneo from Sabah and Sarawak separated approximately 4.5 Ma in the early Pliocene, followed by further cladogenesis in Sarawak through the Pleistocene. Divergence of peninsular Malaysian populations from the Mentawai Archipelago occurred approximately 5 Ma. Separation among island populations from the Mentawai Archipelago likely dates to the Pliocene/Pleistocene boundary approximately 3.5 Ma, and our samples from peninsular Malaysia appear to coalesce in the middle Pleistocene, about 1 Ma. Coupled with the monophyly of these populations, these divergence times suggest that despite consistent land-connections between these regions throughout the Pleistocene E. rugifera still faced barriers to dispersal, which may be a result of environmental shifts that accompanied the sea-level changes.

Introduction

Throughout the Pleistocene and late Pliocene, glaciation cycles caused sea level fluctuations that repeatedly led to land connections between islands that today are separated by ocean channels (Miller et al., 2005). With these land connections, populations of terrestrial organisms that may have started to diverge on the separated landmasses have an opportunity to reconnect. Notable global cases (and examples of investigations of faunal divergence between the landmasses) include the reconnection of Taiwan with mainland Asia (Oshida et al., 2017), Tasmania with Australia (Dubey, Keogh & Shine, 2010), Sri Lanka with India (Bossuyt et al., 2004; Bauer et al., 2010), Japan and the Ryukyus with east Asia (Ota, 1998; Qi et al., 2014) and islands of New Zealand with one another (Greaves et al., 2008). In some cases, divergence between allopatric populations may continue to accumulate despite land connection, and in other cases populations may fully reintegrate into panmixia. The likelihood that the populations of an organism will reconnect or remain divergent is determined by the dispersal capability of the organism across the exposed land-bridge, which is affected by the unique ecologic and geographic forces that exist in that region. These biogeographic drivers are an active area of research in homologous systems affected by sea level fluctuations (e.g., the Philippines, see Brown et al., 2013 for review).

The historic sea level fluctuations on the Sunda Shelf represent a dramatic case of an ever-changing Pleistocene landscape. Sea levels dropped 120 m below present levels and caused land area on the Sunda Shelf to expand up to twice the present area, leading to concomitant climatic and environmental changes (Cannon, Morley & Bush, 2009). Sea levels were consistently lower (on average 62 m below present levels over the past 1 Ma) than today throughout the Pleistocene, during which about 58 glaciation events occurred, usually every 50–100 ka, that allowed for broad land connections between the present-day landmasses (Voris, 2000; Sathiamurthy & Voris, 2006; Woodruff & Turner, 2009). Thus, the forest distributions and island positions and connections on the Sunda Shelf during interglacial periods of high sea levels, such as the present, represent the exception rather than the norm (Cannon, Morley & Bush, 2009). With these broad land connections so consistently bridging the islands, one would predict that for widespread species divergence times between populations across the Sunda Shelf will date back to one of the sea level reductions during the last 2.9 Ma when gene flow would have been likely between the populations (Woodruff & Turner, 2009). This pattern of late-Pleistocene divergence for populations across the Sunda Shelf has been observed in several birds and some reptiles (Lim et al., 2011; Grismer et al., 2015). However, older divergence times predating these Plio-Pleiostocene land connections have been found in several mammals (e.g., (Gorog, Sinaga & Engstrom, 2004; Steiper, 2006; Den Tex et al., 2010), suggesting that barriers may have existed on the exposed Sunda Shelf.

The rough-scaled sun skink, Eutropis rugifera (Stoliczka, 1870), is a secretive, semi-arboreal skink, distributed over mainland Southeast Asia (Peninsular Malaysia and southern Thailand), as well as on the islands of Borneo, Sumatra, Java, Bali, the Nicobar Islands, the Mentawai archipelago, and the southwest Philippines (Das, 2010; see Fig. 1). Barley et al. (2015) and Amarasinghe et al. (2017) reported E. rugifera from Sulawesi in error based on samples from the Mentawai archipelago that Barley et al. incorrectly attributed to Sulawesi. Inhabiting low and mid elevation rainforests as well as peat swamps, E. rugifera is the most arboreal Eutropis on Sundaland and, although generally terrestrial, has also been found up to a height of 2 m on tree trunks (Das, 2004). Eutropis rugifera is also one of the more enigmatic species of Eutropis, with relatively few museum specimens collected and little known of its natural history.

Figure 1 Map of the Sunda Shelf with 40 m and 120 m bathymetric contours.

Land connections on the Sunda Shelf form with a drop in sea level of just 40 m below present levels. During the last glacial maximum (LGM), sea levels in the area dropped to 120 m below present levels. The bold black shape indicates the range of Eutropis rugifera, including the presently recognized Philippines clade. Sampling localities of genetic samples used in this study are depicted by dots, and are color coded to match genetic clades inferred with the molecular analyses (Figs. 2–3). Map generated in QGIS (QGIS Development Team, 2017), with bathymetric contours sourced from the GEBCO Digital Atlas published by the British Oceanographic Data Centre on behalf of IOC and IHO, 2003.

As Eutropis rugifera is present on the four largest Sundaic landmasses (Borneo, Sumatra, Java, and Peninsular Malaysia), it represents an ideal biological system for testing hypotheses of divergence and diversification on the Sunda Shelf. Sea level and climatological shifts have impacted the landscape dramatically. It is estimated that Borneo was connected to the Malay Peninsula throughout the Miocene, and to Java and Sumatra as well after their emergence in the mid-Miocene (∼10–15 Ma). These broad land connections on the Sunda Shelf persisted until the early Pliocene (∼5 Ma) when sea level shifts caused fragmentation of these islands to near-present configurations, though there is likely to have been occasional land connections due to sea-level shifts throughout the Pliocene (Gorog, Sinaga & Engstrom, 2004; Hall, 2009; Woodruff & Turner, 2009). From the mid-Miocene to the mid-Pliocene, the climate is predicted to have remained relatively humid and moist, with tropical rainforest blanketing the Sunda Shelf. However, a global cooling event in the late Pliocene (∼3.3 to 2.5 Ma) caused substantial aridification that likely led to increased grassland on the Sunda Shelf and fragmentation of habitat (Dersch & Stein, 1994; Gorog, Sinaga & Engstrom, 2004; Miller et al., 2005). At the end of the Pliocene and throughout the Pleistocene, the frequency of rapid sea-level shifts increased dramatically, with land-masses on the Sunda Shelf consistently connected during sea-level drops of >40 m (Voris, 2000; Woodruff & Turner, 2009). Despite the consistency of land connections on the Sunda Shelf within the last 3 Ma, there is evidence that during these periods of land connections the central Sunda Shelf was relatively more xeric than at present and comprised chiefly savannah habitat (Bird, Taylor & Hunt, 2005; Cannon, Morley & Bush, 2009) that could have prevented dispersal of rainforest adapted taxa. Molecular analyses in mammals (Gorog, Sinaga & Engstrom, 2004) and birds (Lim et al., 2011) support the presence of barriers to dispersal on the exposed Sunda Shelf, although other evidence suggests a heterogeneous landscape (see Louys & Meijaard, 2010).

Previous phylogenetic studies have included broad enough geographic sampling for E. rugifera to allow for a minimal assessment of phylogeographic patterns, but not to test the drivers of divergence within the species. An initial phylogenetic study of Eutropis found little molecular divergence in E. rugifera across three localities in Indonesia (Mausfeld & Schmitz, 2003). Adding to this dataset, topotypic material from the Nicobar Islands was found to show moderate molecular mitochondrial divergence from Sundaic (Bali and Sumatra) E. rugifera (mean uncorrected p-distance 4.6% for 12S; 2.5% for 16S; Datta-Roy et al., 2012). Amarasinghe et al. (2017) included additional samples from Bali and from Bawean island to the north of Java, uncovering these populations as separate and divergent lineages. Increased divergence is common in island populations, and this level of divergence was not considered substantial enough to distinguish these populations as separate species. Using unique molecular markers and samples, Barley et al. (2015) recovered Sundaic populations (Borneo, Peninsular Malaysia, and from Pulau Siberut, off the west coast of Sumatra) forming a clade that was highly divergent (15.5–16.5% uncorrected p-distance in ND2) from a population in the Zamboanga Peninsula of Mindanao in the Philippines. This result suggests that the population in the Philippines represents a hitherto undescribed species within the group.

In this study, we test whether divergence times between Sundaic populations of E. rugifera correspond to periods of land connections across the Sunda Shelf during the Pleistocene, suggesting that these land bridges allowed for dispersal across the Sunda Shelf, or if they predate the late-Pliocene and Pleistocene climatic shift, indicating that E. rugifera faced barriers to dispersal on the Sunda Shelf despite land connections. The result has implications for the role historical climatic and geographic processes have played in the evolutionary history in Eutropis rugifera and other taxa on the Sunda Shelf.

Materials & Methods

The dataset used for phylogenetic analysis included five protein coding markers: mitochondrial ND2 (1,029 bp), and nuclear BRCA1 (969 bp), BRCA2 (1,227 bp), MC1R (660 bp) and RAG1 (1,131 bp) amplified using polymerase chain reaction. Corresponding primers and annealing temperatures are shown in Table 1. When combined with samples from GenBank, the total dataset included 18 specimens of E. rugifera, with the Philippine samples treated as an outgroup (based on Barley et al., 2015). All sequences are deposited on Genbank (see Table S1 for Genbank accession numbers). Bayesian Inference (BI) was conducted in MrBayes v3.2.1 (Ronquist & Huelsenbeck, 2003) and Maximum Likelihood (ML) analysis was conducted using RAxML v8.1.15 (Stamatakis, 2014). All genes were concatenated and the appropriate partitioning scheme for each analysis was determined using PartitionFinder v1.1.1 (Lanfear et al., 2012) based on the Bayesian Information Criterion. For RAxML, the GTR + Γ model was specified for all partitions specified by PartitionFinder with 1,000 rapid bootstrap replicates to determine nodal support. For MrBayes, the analysis was set for 50,000,000 generations, with 4 chains, and two independent runs. The first 25% of trees were discarded as burn-in from each run, and the latter 75% of trees from both simultaneous runs were combined. For all Bayesian analyses, adequate burn-in and convergence of the Markov chains was assessed by eye using Tracer v1.6 (Rambaut & Drummond, 2013), and all ESS values were confirmed to be greater than 200 (most were greater than 8,000).

Table 1 List of genes, primer names and sequences, references, and annealing temperatures used for each of the genes in this study.

Gene	Primer name	Reference	Primer sequence (5′–3′)	Annealing temp. (°C)	
ND2	MetF1	Macey et al. (1997)	AAGCTTTCGGGCCCATACC	50	
	CO1R1	Arevalo, Davis & Sites (1994)	AGRGTGCCAATGTCTTTGTGRTT		
BRCA1	BRCA1skink1804F	Karin et al. (2016)	YWTGGAGYTGAAYCCAGAAACTGATG	56	
	BRCA1skink3100R	Karin et al. (2016)	RKWGTCCTCAGAYKCATGWGACTGGG		
BRCA2	BRCA2skink984F	Karin et al. (2016)	AACAGGTAGTCAGTTTGAMTTYACAC	56	
	BRCA2skink2315R	Karin et al. (2016)	RTTGAAGYYTGAATGCYAGGTTTGAC		
MC1R	MC1R.F	Pinho et al. (2010)	GGCNGCCATYGTCAAGAACCGGAACC	54	
	MC1R.R	Pinho et al. (2010)	CTCCGRAAGGCRTAAATGATGGGGTCCAC		
RAG1	RAG1skinkF2	Portik, Bauer & Jackman (2010)	TTCAAAGTGAGATCGCTTGAAA	50	
	RAG1skinkR2	Portik, Bauer & Jackman (2010)	AACATCACAGCTTGATGAATGG		
	RAG1skinkF370	Portik, Bauer & Jackman (2010)	GCCAAGGTTTTTAAGATTGACG		
	RAG1skinkR1200	Portik, Bauer & Jackman (2010)	CCCTTCTTCTTTCTCAGCAAAA		

Divergence times were estimated on the 5-loci dataset using the program BEAST v1.8.2 (Drummond & Rambaut, 2007). The results of PartitionFinder were modified to allow for separate substitution rate estimations for each locus. This resulted in separate partitions for the first, second, and third positions for ND2 and for each nuclear marker in separate partitions with the first and second codon positions together, and the third codon position separate, for a total of 11 partitions. All nuclear markers were run under the HKY  + Γ model, and ND2 was run using the TrN + Γ model as specified in PartitionFinder for BEAST. The priors were set as follows: coalescent, constant population size tree prior; lognormal prior with standard deviation of 1 on each of the relative codon rate priors. A strict clock rate was chosen for all of the markers, with the substitution rate for the nuclear markers estimated relative to ND2. This was achieved by placing a flexible uniform prior (from 0 to 1) on the nuclear substitution rates, and by sampling the ND2 rates from a normal distribution with a mean of 0.00895 and a standard deviation of 0.0025 (which spans the ND2 rates observed in many taxa; Barley et al., 2015). The Markov chain was run for 50 million generations, and convergence and appropriate burn-in was assessed in Tracer, as specified above.

Haplotype networks of the nuclear markers were generated in R (R Core Team, 2016) using the pegas package (Paradis, 2010). Samples were organized based on locality, with increased subdivisions in northern Borneo. Philippine samples were excluded from the analyses. Networks were generated under a parsimony model, and alternative topologies are displayed by dashed grey lines.

Recently collected samples from Sarawak were collected and exported under permits approved by the Sarawak Forestry Department (Research Permit NCCD.907.4.4(Jld.11)-68 and Export Permit 15530). Collection and euthenization protocols were approved by the Villanova University Institutional Animal Care and Use Committee (AS FY13-14 and AS FY17-02).

Results

The concatenated BI and ML topologies were nearly identical, except for one node within the north Borneo clade (Fig. S1). Support values differed substantially, with the BI analysis showing higher posterior probabilities. All analyses supported E. rugifera from Sundaland as monophyletic and divergent from populations from Mindanao in the Philippines, as previously reported by Barley et al. (2015). In all analyses, individuals from north Borneo represent a well-supported clade. The BI and ML analyses supports the placement of the Peninsular Malaysian samples as more closely related to Bornean samples and sister to the Mentawai archipelago samples (Fig. S1). The topology in the timetree, however, places Peninsular Malaysian samples sister to Mentawai Archipelago samples (Fig. 2).

Figure 2 Divergence time estimates for Eutropis rugifera from analysis with BEAST.

Node labels show median node ages, with confidence intervals depicted by blue bars. Posterior probabilities greater than or equal to 0.95 designated by a closed circle at the node. The bold arrow shows the node of particular interest for the divergence of the Sundaic groups, and suggests divergence time across the Sunda Shelf that predates the Plio-Pleistocene transition. The “Climate” time-bar depicts the historical transition (gradient color, dashed gray bar above) from the mesic Miocene and Pliocene into the more xeric Pleistocene (see text for explanation). The “Land Connections” time-bar depicts the periods of time with sea level at least 40 m below present levels, when all major land-masses on the Sunda Shelf were connected (Miller et al., 2005). The frequency of the sea level shifts increased dramatically at the end of the Pliocene and throughout the Pleistocene.

There exist three divergent lineages of E. rugifera in north Borneo, one in central Sarawak from Bintulu to Gunung Mulu National Park, one in far southwest Sarawak, in Kuching, and a third in Sabah. The relationship between these clades is not well supported, but is consistent among all analyses in showing the sample from Kuching sister to the central Sarawak clade, and the sample from Sabah sister to all the Borneo samples. Divergence time estimates place the split between these clades at well over 2 Ma. The BI analysis places the Kuching and the Danum Valley specimens together as sister to the central Sarawak clade. The BI and ML topologies do not recover strong support for these relationships, but the timetree shows some support for these relationships (Fig. 2).

Divergence time estimates based on a strict molecular clock (Fig. 2) relative to the mitochondrial marker show the 95% confidence interval of divergence between E. rugifera in Sundaland and E. cf. rugifera in the Philippines to be 9.87–24.63 Ma. The crown Sundaland clade dates back to 3.49–8.36 Ma, which predates the Pleistocene land-connections on the Sunda Shelf. The crown node of the Bornean clade is dated to 3.49–8.36 Ma, and the crown node for the split between the Peninsular Malaysia and Mentawai populations is at 3.18–7.57 Ma.

Haplotype networks show the varying levels of phylogenetic signal among the four nuclear genes (Fig. 3), with BRCA1 containing the highest genetic diversity, followed by RAG1, BRCA2, and MC1R. BRCA1 and RAG1 are consistent in showing similar haplotypes among Peninsular Malaysian and Mentawai Archipelago samples. BRCA2 and MC1R show lower levels of differentiation, with samples from across the range sharing MC1R haplotypes. Nuclear trees are not discordant with patterns shown in the concatenated tree, in general showing Peninsular Malaysian and Mentawai Archipelago with divergent haplotypes from north Borneo samples (also see Fig. S2).

Figure 3 Haplotype networks of Sundaic E. rugifera for four nuclear genes colored to correspond to the major clades and geographic locations in the concatenated analyses.

Haplotype circles are scaled to the number of specimens showing a haplotype. The number of hash-marks between circles denotes the number of nucleotide changes.

Discussion

Our molecular clock dating estimates in E. rugifera, place the crown node for the Sunda clade between 3.49–8.36 Ma, and this entire confidence interval predates the period of consistent land connections on the Sunda Shelf (see Fig. 2). Monophyly of north Borneo and Peninsular Malaysian samples further supports the hypothesis that these island populations have remained in isolation through the Pleistocene land connections, and deep divergences within north Borneo samples suggest Pleistocene barriers may have existed not only on the Sunda Shelf, but on Borneo as well. We cannot differentiate between mechanisms causing the initial cladogenesis among populations on the Sunda Shelf without a time-calibrated tree with narrower confidence intervals. Possible scenarios include vicariance during Pliocene fragmentation of the Sunda shelf (Gorog, Sinaga & Engstrom, 2004), oceanic dispersal between islands during Pliocene periods of high sea-levels, or population fragmentation during the Plio-Pleistocene climatic shift.

Ecological and physical barriers on the exposed Sunda Shelf during the Pleistocene may have prevented gene flow across it for low and mid elevation rainforest adapted species. Multiple lines of evidence suggest that the when exposed, the central Sunda Shelf supported an open steppe habitat of grassland and savannah with mangrove forests and swamplands near the coast (Gathorne-Hardy et al., 2002; Bird, Taylor & Hunt, 2005; Cannon, Morley & Bush, 2009; Louys & Meijaard, 2010). Environmental models, however, have difficulty determining if an “arid corridor” that would have completely separated LGM lowland evergreen rainforest on Borneo from that on Peninsular Malaysian and Sumatra existed (Cannon, Morley & Bush, 2009). Our phylogeographic results are consistent with the hypothesis that the central core of the Sunda Shelf was not suitable for E. rugifera, and gene flow was extremely limited even during periods of Pleistocene land connections. As a rainforest adapted species, we suspect that E. rugifera was unable to cross the exposed Sunda Shelf due to ecological unsuitability in this area, resulting in divergence times that predate the Pleistocene (Fig. 2). Unfortunately, locality records of E. rugifera are limited and geographically clustered, many of which are not georeferenced accurately, and therefore do not allow for adequate ecological niche models that can be projected on the paleoclimate and which would allow for a more thorough understanding of the species’ response to the historical conditions (Lim et al., 2011). Alternatively, if E. rugifera did not face ecological barriers on the Sunda Shelf, then the incumbency of allopatric E. rugifera in the same environmental niche could also have prevented panmixia (Kozak & Wiens, 2006). This is possible if some level of reproductive isolation had evolved between populations on different Sundaic landmasses, or if population sizes were large and/or if gene flow was limited temporally and in magnitude during these periods.

Within the Borneo clade, we find some evidence of a population break between Sarawak and Sabah, as well as between central Sarawak and western Sarawak. Genetic divergence between populations in Sabah and Sarawak has been observed in many bird species (Moyle et al., 2005; Sheldon et al., 2009; Hosner et al., 2010; Lim & Sheldon, 2011; Lim et al., 2011; Moltesen et al., 2012; Den Tex & Leonard, 2013; Gawin et al., 2014), in some frogs (Brown et al., 2009; Brown et al., 2010; Arifin et al., 2011; Brown & Siler, 2014), tree squirrels (Den Tex et al., 2010), honeybees (Tanaka et al., 2001), and in some trees (Kamiya et al., 2002; Ohtani et al., 2013); however, it remains unknown what biogeographic barrier or historical process is causing this divergence. In some cases, populations in western Sarawak are resolved as sister to populations in Sumatra or Peninsular Malaysia (Lim et al., 2011), suggesting that these species were able to cross the exposed Sunda Shelf, but faced stronger barriers to gene flow within Borneo. Our results suggest that E. rugifera not only was unable to cross the Sunda Shelf during periods of reduced sea-levels, but also faced strong barriers to gene flow within Borneo, as evidenced by the deep divergences observed across northern Borneo.

Our phylogeographic results are consistent with Amarasinghe et al. (2017), who recovered substantially divergent lineages on Bali and Bawean island (although they did not estimate divergence times). If we were to combine the results of Amarasinghe et al. (2017) with our own (different genetic loci), it is possible that there are between three and six major clades of Sundaic E. rugifera (and potentially even more from unsampled localities): (1.) Sumatra + Nicobar Islands; (2.) Mentawai Archipelago; (3.) Peninsular Malaysia; (4.) Borneo; (5.) Bawean Island; and (6.) Bali. Further genetic sampling may show close relationships between some of these clades (e.g., Mentawai Archipelago closely related to Sumatra and the Nicobar Islands; or Bawean Island closely related to the Borneo clade) and will shed light on the phylogeography and evolutionary history of this species across the entire Sunda Shelf.

Modern herpetological collections in Indonesia have been sparse and large portions of the country remain to be surveyed. Secretive species like E. rugifera, which are relatively difficult to target when conducting fieldwork, will only be fully understood once a larger body of genetic samples have accumulated from repeated surveys across Indonesia. In particular, samples from Kalimantan are necessary to investigate the geographic structuring observed in Borneo, and collections across the entire range of E. rugifera, especially from the Nicobar Islands, Sumatra, and Java will help to fully understand the phylogeography of this species. The divergent and geographically isolated lineages of E. rugifera uncovered here both within Borneo and across the Sunda Shelf suggest that this species may be particular susceptible to divergence from biogeographic forces, and further study of this species may lead to further insights into the evolutionary processes causing cladogenesis on the Sunda Shelf.

Supplemental Information

Figure S1 The concatenated BI phylogeny with support displayed for BI/ML analyses

Support for the ML phylogeny indicated by bootstrap values (proportion of 1,000) and for the BI tree by posterior probabilities.

Click here for additional data file.

Figure S2 RAxML gene trees with 100 rapid bootstrap replicates

Click here for additional data file.

Table S1 Specimens, localities, and associated genbank accession numbers included in the study

Click here for additional data file.

Supplemental Information 1 Sequence data

Click here for additional data file.

We thank Haji Wan Shardini Wan Salleh, Engkamat Lading, Sapuan Ahmad and the Sarawak Forestry Department for facilitating collecting (NCCD.907.4.4(Jld.11)-68) and export (No. 15530) permits. We thank graduate students Adi Shabrani and Pui Yong Min (Institute of Biodiversity and Environmental Conservation, Universiti Malaysia Sarawak) and Jackie Childers and Ian Brennan (Villanova University) for help in the field. Finally, we thank Jens Vindum and Lauren Scheinberg (CAS) for assistance with the importation of specimens.

Additional Information and Declarations

Competing Interests

Author Contributions

Animal Ethics

Field Study Permissions

DNA Deposition

The authors declare there are no competing interests.

Benjamin R. Karin conceived and designed the experiments, performed the experiments, analyzed the data, wrote the paper, prepared figures and/or tables, reviewed drafts of the paper.

Indraneil Das, Todd R. Jackman and Aaron M. Bauer conceived and designed the experiments, contributed reagents/materials/analysis tools, reviewed drafts of the paper.

The following information was supplied relating to ethical approvals (i.e., approving body and any reference numbers):

Collection and euthanasia protocols were approved by the Villanova University Institutional Animal Care and Use Committee (AS FY13-14 and AS FY17-02).

The following information was supplied relating to field study approvals (i.e., approving body and any reference numbers):

Collecting and export permits for this study were approved by the Sarawak Forestry Department.

The following information was supplied regarding the deposition of DNA sequences:

Sequences are available at Genbank (Table S1).

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
