# Peer review of "Ancient divergence time estimates in Eutropis rugifera support the existence of Pleistocene barriers on the exposed Sunda Shelf"

_PeerJ, doi:10.7717/peerj.3762_

## Round 0.1 · original submission · Minor Revisions

· Academic Editor

Minor Revisions

The two reviewers have provided positive and detailed comments on your submission. I agree that the data merit publication in PeerJ, but presentation should be improved in a number of ways as suggested by the reviewers.

The main interest of this paper lies first in the fundamental phylogeographic question of repeated vicariant sundering, second in the geographic location, and only third in the particular taxa used. Start the paper with some general global examples (see rev1), then talk about Sunda specifically (Fig1; possibly use text 210-224 as suggested by rev2), then finally say something briefly about the skinks.

Both reviewers find the MaxEnt/Fig5 analysis largely superfluous. I agree that it doesn't add much to your paper in the light of sparse sampling. I also suggest getting rid of Fig2 since it doesn't add anything to Fig3.

Both reviewers make several helpful suggestions to improve Figure labelling, which should be followed.

Rev2 makes some helpful suggestions on terminology, to which I would add a plea to use the construction: "to test whether" rather than "to test if", which appears repeatedly.

Rev1 makes some suggestions about analyses (alternatives to concatenation and other models) which you should consider.

·

Basic reporting

The quality of written communication is excellent. There is a wide literature base employed, although literature from other homologous study systems (land faunas that were periodically connected via Pleistocene sea level fluctuations) was not employed. Many examples exist (Taiwan, Japan, Tasmania, New Zealand, Sri Lanka, England).

Tables and Figures were all of a high standard, although Fig1 needed the locality names on the figure for each sample, to avoid the need of cross-referencing with other sources. Likewise, Figure S1 only had codes for samples, and hence was uninformative without consulting other information. These figures need to be "stand alone" with respect to interpretation.

One of the main deficiencies was that the Introduction started straight with the study species. The question is very interesting and broadly relevant, so the paper should start with the over-arching issue, that is of broadest interest and significance (genetic history of taxa on landbridge islands, and predicting their past connectivity using SDMs as verified by genetics).

Experimental design

It was unfortunate that the previously studied specimens and DNA sequences could not be integrated into this study. As such, it comes across as the third study to make a "first attempt" at the phylogeography of this species.

In terms of phylogenetic reconstruction, concatenation of genes is only one option. The alternative, that is more realistic, is to build population trees that allow each gene to have evolved along an independent gene tree. There are methods out there that do this (including BEAST). That said, it seems most of your variation is harboured in mtDNA, and the nDNA are not adding much.

I was not sure about your models employed during tree building. I cannot see why the BEAST strategies needed to be different from MrBayes and RAXml, and vice versa (particularly with respect to codon models). Even if you need to fragment your datafile, you can make datasets of 1st codons all together, all 2nd codons together, and so forth, and then give them separate models in MrBayes or RAXml

Given the depth of divergences at play, is a coalescent tree prior likely to be the most appropriate for BEAST analysis, versus a Yule tree prior?

I am not very familiar with MaxEnt, but I am familiar enough to know that a basic acceptance of default parameters is inappropriate, and you should find good, more rigorous examples of its implementation to follow as a guide. This will indicate the degree of sensitivity to parameter selection.

No outgroup in the analyses? Although the Philippine split looks obvious, without an outgroup, you actually cannot say where the deepest split (phylogentically) occurs.

Validity of the findings

The findings are probably valid, but rely heavily on the mtDNA variation, which should be more adequately explored. Otherwise, the results may be given the impression that they are more reliable (strong multilocus support) than reality.

One of the big issues I have is an apparent contradiction. The species could not move across the Sunda shelf, to mix with other populations. However, the SDMs should that they also did not persist at most of the contemporary sample sites during the Pleistocene glaciations. So, if they couldn't move widely, and they could not persist where they are today, how did they come to have their present distribution, and indeed, deep genetic divergences?

I would be interested to know whether there are any modern day homologs of the glacial Sunda shelf habitat, and whether the species occurs there today or not. Essentially, is the Sunda shelf glacial habitat novel, relative to today's environment? If so, we have no evidence (from present species distributions) that they could not occupy this historical habitat, because there are no modern homologs.

Comments for the author

The paper seems to end strongly on the point "we need more sampling". This weakens the study, because it points out your main deficiency, and leads to the question, why didn't you do more sampling?

Reviewer 2 ·

Basic reporting

My main suggestions for revision are in "General Comments for the Author" below. Included in these comments are suggestions for improving the focus of the text on the major discoveries made possible by the new data. Much of what exists in this draft seems largely extraneous to this central purpose.

Experimental design

I have no major changes to suggest on this topic.

Validity of the findings

Validity seems fine to me.

Comments for the author

This manuscript reports new comparative DNA sequence data and phylogenetic analyses for skink populations of the species Eutropis rugifera. Geographic sampling is broad but sporadic, providing a historical biogeographic overview for this species. The results are worth publishing, but the manuscript needs a reasonable amount of revision to be effective in highlighting the new findings.

The major problems with the presentation are evident from the abstract, which should describe the new data (genes, number of populations sampled) and analyses, and the major conclusions. For the latter, a verbal account of the contents of Figure 3 would be a good strategy: “The earliest phylogenetic divergence within E. rugifera separates Philippine populations from those of the Sunda Shelf with an estimated divergence of 16 million years ago; the Philippine populations thus cannot be considered conspecific with their Sunda congeners. Within the Sunda Shelf, the oldest divergence separated Borneo populations from a common ancestor of populations in peninsular Malaysia and the Mentawai Archipelago approximately 6 million years ago. Our data suggest that within Borneo, populations from Sabah and Sarawak separated approximately 4-5 million years ago in the early Pliocene, followed by further cladogenesis in Sarawak through the Pleistocene. Separation of continental Malaysian populations from those in the Mentawai Archipelago occurred approximately 5 million years ago. Divergence among samples from the Mentawai Archipelago likely dates to the Pliocene/Pleistocene boundary approximately 3.5 million years ago, and our samples from continental Malaysia appear to coalesce in the middle Pleistocene, about 1.5 million years ago.” With this temporal and spatial framework, focus the historical biogeographic analysis and discussion specifically on these events. Given the full temporal scope of the divergences revealed within E. rugifera, the emphasis on the Pleistocene in the abstract and elsewhere is misplaced; only the most recent divergences within Sarawak and coalescence of the continental Malaysian samples date to the Pleistocene, so these are the only events for which Pleistocene climatic history is particularly relevant. Limit your discussion of the historical climate of the Sunda Shelf to what should be your main point: the most parsimonious explanation of the historical events revealed by your data analysis. Your bar on Figure 3 showing episodic land connections probably suffices for this purpose. Extensive discussion of grasslands and rivers in the past history of the Sunda Shelf seems unwarranted by the data; introduce these considerations only if necessary to explain a specific case of cladogenesis on Figure 3.

Some additional problems in the abstract also recur throughout the manuscript. “Last Glacial Maxima” is a misnomer. The correct phrase is “Last Glacial Maximum,” the single most recent time that glaciation reached a maximum southern extension from the northern pole. The LGM occurred about 10,000 years ago, much too recent to be of explanatory value for the divergence events revealed by your data. Statements such as “unexpectedly ancient divergence” in line 26 are too vague to be informative; substitute the specific numerical estimates from your data and analyses for “ancient,” “recent,” “close” or “distant” in describing divergences, as illustrated in my suggested rewording of your main discoveries. In describing recurring cycles of inundation versus exposure of Sunda land areas, I favor “episodic” rather than “periodic,” the latter being a special case of “episodic” in which successive episodes have a constant temporal spacing. Figure 3 indicates that perhaps the Pleistocene cycles approach periodicity, but “episodic” is the more conservative claim.

One important concept is missing from your discussion of historical biogeography: “incumbency” of populations in their interaction upon secondary contact. Vicariant populations of E. rugifera that have evolved as distinct lineages over many millions of years likely have made secondary geographic contact episodically following their initial divergence as climatic conditions temporarily removed a geographic barrier (for example, a drop in sea level permitting formerly separated populations to make contact by expanding into a newly formed land bridge between them). Because of their ecological similarities, each population lineage likely blocked the spreading of its geographic neighbor into its own range at these times of transitory secondary contact. The geographic pattern of vicariance thus persisted over many millions of years despite episodic removal of the physical barrier that initially led to allopatric speciation. Many authors have discussed this phenomenon, perhaps most notably John Wiens and Kenneth Kozak, although my recollection is that the term “incumbency” traces to writings by Elizabeth Jockusch and David Wake on Batrachoseps salamanders. Given your sparse geographic sampling, detailed discussion of incumbency is not advised here; the phenomenon nonetheless allows you to focus your explanations on what caused the initial cladogenesis of evolutionary lineages without having to explain why they subsequently maintained their patterns of vicariance despite episodic geographic contact during the Pleistocene.

I focus next on the figures, which mostly succeed in conveying the main new discoveries, but which could be improved.

Figure 1 – Label all of the main areas discussed in your text. The Mentawai Archipelago is the most obvious omission; move “Eutropis rugifera Range” to the top of the circled area to make room to label the Mentawai Archipelago where “Eutropis rugifera Range” now appears. Within Borneo, distinguish Sabah and Sarawak. Because you mention them in lines 235-236, label Nicobar Islands, Bali and Bawean Island. Also, number the localities and use the same numbers to cross-reference taxa in Figures 2 and 3.

Figure 2 – After numbering the geographic localities shown on Figure 1, include the numbers in the tip taxa for cross-reference.

Figure 3 – Again, add numbers to the tip taxa to reference Figure 1 as requested also for Figure 2. Because the main purpose of this figure is to convey the temporal information, remove Bayesian support from the nodes and replace it with the estimated time of divergence. If you want to convey branch support, then perhaps put a small black circle at each node whose Bayesian posterior probability is 0.95 or greater. Then your numbers and confidence intervals on the figure will be synchronized. Because only one event of cladogenesis coincides with the transition from mesic to xeric climate at the late Pliocene, this point is perhaps overemphasized. The overall importance of this transition is to explain why episodes of land connection are much more frequent following this point, although your most important events of cladogenesis precede it.

Figure 5 – Perhaps cut this analysis entirely. My earlier comments explain why I consider it largely irrelevant to your data. You want to have some hypotheses of the geographic barriers that explain the initial cladogenesis of your main lineages. It is potentially of interest to identify what areas of refuge allowed these lineages to persist when unfavorable climate restricted their distributions; nonetheless, your geographic sampling is too sparse for giving too much emphasis to this point. Perhaps show one map that depicts the areas most likely to have been continuously occupied by lineages of E. rugifera through the cycles of fluctuating climate. This is the main point that your readers will want to know. The current presentation is too complicated to convey this point effectively.

From my preceding comments, it is evident that I would greatly condense and rewrite the introduction and discussion sections to focus on explaining the main new results. Material on lines 210-224 of the Discussion should be consolidated with the introduction. Include at the start of the Results section a formal description of your new data before covering results of the phylogenetic analyses of those data. Organize your discussion section chronologically by summarizing the main divergence events and their best historical geographic explanations from the deepest to most recent cladogenetic events. As you reach the tips of the tree, identify the minimum number of separate geographic lineages that your data have revealed.

Lines 267-271 are mostly a distraction. Where internal branches on a phylogenetic tree are small enough that topology cannot be resolved as successive dichotomies, emphasize instead the approximate time at which the various branches coalesce. Especially for cases of vicariance caused by interaction of a global climatic feature with details of topography, it is likely that some phylogenetic splits will be true polytomies rather than successive dichotomies. In such cases, the relevant phylogenetic resolution consists in putting a precise time estimate on the polytomy, not in trying to force it into a dichotomous scheme. Your main message then should be that the various lineages arose approximately simultaneously at the estimated divergence time that best fits your data.

---

## Round 0.2 · accepted · Accept

· Academic Editor

Accept

You have followed the recommendations closely, and the manuscript looks much better. I just notice that LGM is still not corrected in the Fig1 legend- it should be "maximum" (singular).